# Antitumor Effects of PRIMA-1 and PRIMA-1^Met^ (APR246) in Hematological Malignancies: Still a Mutant P53-Dependent Affair?

**DOI:** 10.3390/cells10010098

**Published:** 2021-01-07

**Authors:** Paola Menichini, Paola Monti, Andrea Speciale, Giovanna Cutrona, Serena Matis, Franco Fais, Elisa Taiana, Antonino Neri, Riccardo Bomben, Massimo Gentile, Valter Gattei, Manlio Ferrarini, Fortunato Morabito, Gilberto Fronza

**Affiliations:** 1Mutagenesis and Cancer Prevention Unit, IRCCS Ospedale Policlinico San Martino, 16132 Genoa, Italy; paola.monti@hsanmartino.it (P.M.); andrea.speciale@hsanmartino.it (A.S.); gilberto.fronza@hsanmartino.it (G.F.); 2Molecular Pathology Unit, IRCCS Ospedale Policlinico San Martino, 16132 Genoa, Italy; giovanna.cutrona@hsanmartino.it (G.C.); serena.matis@hsanmartino.it (S.M.); franco.fais@unige.it (F.F.); 3Department of Experimental Medicine, University of Genoa, 16132 Genoa, Italy; ferrarini.manlio@gmail.com; 4Department of Oncology and Hemato-Oncology, University of Milan, 20122 Milan, Italy; elisataiana@gmail.com (E.T.); antonino.neri@unimi.it (A.N.); 5Hematology Unit, Fondazione IRCCS Ca’ Granda, Ospedale Maggiore Policlinico, 20122 Milan, Italy; 6Clinical and Experimental Onco-Haematology Unit, Centro di Riferimento Oncologico, IRCCS, 33081 Aviano, Italy; rbomben@cro.it (R.B.); vgattei@cro.it (V.G.); 7Unità di Ricerca Biotecnologica, Azienda Sanitaria Provinciale di Cosenza, 87051 Aprigliano, Italy; m.gentile@aocs.it (M.G.); f.morabito53@gmail.com (F.M.); 8Department of Hematology and Bone Marrow Transplant Unit, Augusta Victoria Hospital, Jerusalem 91191, Israel

**Keywords:** PRIMA-1/APR246, hematological malignancies, mutant P53, CLL, leukemia

## Abstract

Because of its role in the regulation of the cell cycle, DNA damage response, apoptosis, DNA repair, cell migration, autophagy, and cell metabolism, the *TP53* tumor suppressor gene is a key player for cellular homeostasis. *TP53* gene is mutated in more than 50% of human cancers, although its overall dysfunction may be even more frequent. *TP53* mutations are detected in a lower percentage of hematological malignancies compared to solid tumors, but their frequency generally increases with disease progression, generating adverse effects such as resistance to chemotherapy. Due to the crucial role of P53 in therapy response, several molecules have been developed to re-establish the wild-type P53 function to mutant P53. PRIMA-1 and its methylated form PRIMA-1^Met^ (also named APR246) are capable of restoring the wild-type conformation to mutant P53 and inducing apoptosis in cancer cells; however, they also possess mutant P53-independent properties. This review presents the activities of PRIMA-1 and PRIMA-1^Met^/APR246 and describes their potential use in hematological malignancies.

## 1. Introduction

The development of personalized therapies for cancer treatment has received considerable attention in recent years. Two main steps are mandatory to achieve an effective personalized therapy: first, the identification of a suitable target involved in one or several pathway(s) that is (are) impaired in a given cancer or in a group of cancers and, second, the identification of a drug that can specifically hit that target, thus reducing the undesired effects of its dysfunction.

One of the targets that best suites the first requirement is the P53 tumor suppressor protein, which is a key player in different pathways ranging from cell cycle regulation, DNA damage response, apoptosis, senescence, DNA repair and cell migration to those more recently identified, such as autophagy and metabolism. Under normal conditions, the intracellular P53 protein is kept at a low level by a negative-feedback loop with the MDM2 E3 ubiquitin ligase, which promotes its degradation in the absence of stress [1]. However, following different types of stress, P53 becomes activated, accumulates in the cell nuclei, and trans-activates a plethora of target genes that build up an appropriate cell response. To achieve these goals, P53 binds, as a tetrameric transcription factor, to conserved response elements (REs) located in the promoter regions of target genes, including *P21* and *GADD45A* for the cell cycle arrest, *BAX*, *PUMA* and *NOXA* for the induction of apoptosis, thus activating their expression and counteracting cancer onset and progression [2,3].

The occurrence of a *TP53* mutation may have important consequences on cell homeostasis: (i) the genes that are normally modulated by wild-type P53 are no longer activated, (ii) the mutant P53 protein is not able to participate in the negative-feedback loop with MDM2, leading to an accumulation of mutant P53 protein itself; (iii) the mutant protein can acquire new functions, the so-called gain of function (GOF) properties [4]. In general, these P53 dysfunctions represent detrimental events in cancer as they facilitate cell growth and prevent the response to therapy.

More than 50% of human tumors [5] show *TP53* mutations, although their frequency varies among different types of tumor ranging from 90% in ovary cancer, 50–80% in lung cancer, 40–60% in colorectal cancer to 10% in prostate cancer. The frequency of *TP53* mutations is also highly heterogeneous in the different types of leukemia and lymphomas, being usually low at diagnosis and reaching values up to 60% at progression or at relapse [5]. Besides mutations, P53 dysfunctions may also be generated by the deletion of the *TP53* locus on the short arm of chromosome 17 [indicated as del(17p)] [6,7] as well as by the abnormally high expression of wild-type P53 negative regulators (e.g., MDM2/4) [8].

Because of its widespread functions in cancer origin and progression, the P53 protein has been considered a promising target for the development of new anticancer strategies. Nowadays, three main strategies are available, including (i) the restoration of wild-type P53 activity by inhibiting its binding with MDM2/4 proteins [9,10]; (ii) the reactivation of the wild-type function in the mutant P53 protein [11] and (iii) the elimination of the undesired GOF activities through degradation of the mutant P53 protein [12,13,14,15].

Based on the above-described general strategies, several molecules have been developed and tested in different tumor models. The appealing feature of these molecules is represented by their ability of inducing an efficient apoptotic response in cancer cells that are generally refractory to chemotherapy. Molecules, such as Nutlin and RITA, inhibit the binding of wild-type P53 to MDM2 and, although with different mechanisms, prevent the negative control of MDM2; this allows the P53 stabilization and the induction of a P53-dependent transcription of genes, which induce cell death [16,17]. Conversely, PRIMA-1 and its methylated derivative PRIMA-1^Met^ (from here on mentioned as APR246) have been isolated as small molecular weight molecules able to restore the DNA binding capacity of different mutant P53 proteins and to induce significant apoptosis in cancer cells carrying a mutant P53 protein [18]. The other interesting approach that may prevent the GOF activities of the mutant P53 and the chemoresistance that is consequent to it is based on mutant P53 deprivation. In cancer cells, mutant P53 is stabilized by the over-expression of the HSP90 protein [19]; therefore, several HSP90 inhibitors, such as 17-AAG or Ganetespib, have been tested for their ability to cause the degradation of mutant P53 and for their capacity of acting as anticancer molecules [20,21]. In addition, HDAC inhibitors, such as SAHA, can induce the degradation of mutant P53 and restrain the growth of mutant P53 expressing tumors [15,20]. Some of these molecules (i.e., nutlins, APR246, ganetespid) have reached an advanced level of preclinical experimentation, and clinical trials have been started [10,11,12].

Despite the promising results obtained with the above-mentioned molecules, APR246 is the only mutant P53-reactivating molecule that has reached an advanced stage of clinical investigation. Initially isolated as a molecule with mutant P53-dependent activities, APR246 is now receiving attention also for its mutant P53-independent effects that can be exploited to sensitize different tumors to therapy. The aim of this review is to present the properties of PRIMA-1 and APR246 and to describe their activities reported in hematological malignancies.

## 2. *TP53* Mutations in Hematological Malignancies

The frequency of *TP53* mutations in hematological cancers is lower than in solid tumors (http://p53.free/.fr). In lymphomas, the incidence of *TP53* mutations varies significantly according to the histological subtype and also to the disease stage; generally, *TP53* mutations are relatively infrequent in low-grade non-Hodgkin lymphoma (NHL), whereas a higher incidence is reported in aggressive NHL subtypes [22]. Burkitt lymphoma (BL) has consistently been found to have a frequency of *TP53* mutations up to 33% of cases, as detected by the sequencing of exons 5–9 of the *TP53* gene [23]. *TP53* is mutated in 3–8% of the acute myeloid leukemia (AML) cases [24], in less than 3% of the acute lymphoblastic leukemia (ALL) cases [25], and in 10–12% of multiple myeloma (MM) cases [26,27]. As a general rule, *TP53* mutations in hematological malignancies are associated with a more aggressive disease course, resistance to therapies, and a dire outcome.

CLL represents approximately 30% of all adult leukemias [28] and is characterized by a clonal expansion of small, relatively monomorphic circulating B cells, which may infiltrate the bone marrow, the lymph nodes, and the spleen [29]. The clinical course is heterogeneous, ranging from a rapid disease progression, requiring early treatment, to decades of survival with minimal or no treatment [30,31].

More than 80% of CLL cases have genomic aberrations at diagnosis, the most frequent being partial deletions at 13q (~55%), 11q (~15%), 17p (~8%), and gain of chromosome 12 (~15%) [32]. The incidence of *TP53* mutations is approximately 5–7% at diagnosis [33,34], but it rises as the disease progresses, reaching approximately 40% in refractory CLL [35,36,37,38]. In general, global P53 dysfunctions [i.e., *TP53* mutations or (del 17p)] are associated with adverse outcomes due to the development of resistance to chemotherapy and chemo-immunotherapies. Therefore, their presence represents a key biomarker that guides the therapy decision [39]. Accordingly, the European Research Initiative on CLL (ERIC) group considers mandatory *TP53* mutational screening for all patients before the starting of any therapy [40]. This approach prevents the use of chemo-immunotherapy in favor of new therapies with BCR (B-Cell Receptor) inhibitors [41,42,43,44,45,46] or BCL-2 inhibitors [47,48] in patients with P53 dysfunctions [49]. In addition, a reassessment of *TP53* status before the initiation of any subsequent line of therapy is recommended to exclude the occurrence/expansion of mutant P53 clones during disease progression [33]. Finally, it should be pointed out that the presence of a P53 dysfunction determined as *TP53* mutation and/or del(17p) has a negative effect on the overall survival of patients treated with the new targeted therapies [50].

The presence of a P53 dysfunction may also influence CLL progression from the early disease stages, measured as time to first treatment (TTFT). In CLL, a large overlap between the presence of del(17p) and a *TP53* mutation at the single patient level is present since the majority of patients present both alterations. These patients generally have a shorter TTFT than patients with no *TP53* dysfunctions (i.e., mutations and/or deletions) [51]. Due to few patients with exclusively del(17p) in the various cohorts investigated, it has not been possible to determine whether the presence of the sole del(17p) confers a worse prognosis than the presence of both del(17p) and a *TP53* mutation; moreover, a note of caution should be suggested by the observation made by different groups that a number of patients with exclusively *TP53* mutations are characterized by the presence of IGHV mutations, which notoriously confers a good prognosis [52]. Thus, it is possible that this group of patients with mutated IGHV genes and *TP53* mutations represents a special subset of CLL that have the tendency to not progress since they are not present in cohorts of patients at later stages of the disease [52]. Very recently, the influence of *TP53* mutations on early disease progression has been investigated in CLL [51]. Since the presence of a *TP53* mutation does not necessarily imply a complete P53 inactivation [34], functional characterization of mutant P53 protein encoded by *TP53* mutations was also performed by using the O-CLL1 observational study (clinicaltrial.gov identifier NCT00917540) that recruited a cohort of clinically and molecularly well-characterized Binet stage A patients. The analyzed mutant P53 proteins appeared to be functionally heterogeneous, but such heterogeneity was not associated with differences in TTFT within the group of patients carrying only the *TP53* mutation without del(17p). Even though this study demonstrates that the occurrence of del(17p) significantly predicts TTFT, while that of a *TP53* mutation alone is unable of such prediction [51], further analysis in a larger cohort is needed to get insights on the impact of the functional heterogeneity of P53 mutants on CLL prognosis and therapy.

## 3. PRIMA-1 and APR246: The First 10 Years of History

PRIMA-1 was identified in 2002 as a molecule able to suppress the growth of human tumor cells in a mutant P53-dependent manner [18]. The tumor suppressor effect of PRIMA-1 was related to apoptosis induction, which indeed was abolished in the presence of caspase inhibitors [18]. Even more striking were the observations related to the capacity of PRIMA-1 of restoring the wild-type P53 conformation and causing the P53 binding to its appropriate target genes in human cell lines expressing different mutant P53 proteins. This demonstration was obtained mainly by using the conformation-specific monoclonal antibodies PAb1620 and PAb240 together with band shift assays. Moreover, PRIMA-1 was shown to induce the expression of *P21* and *MDM2* target genes in H1299 cells (a non-small cell lung carcinoma cell line expressing the R175H mutant), and in SW480 cells (a colon adenocarcinoma cell line carrying the endogenous R273H mutant). In vivo experiments in human tumor cells xenografted in mice revealed that the antitumor effect of PRIMA-1 was dependent on mutant P53 expression [18]. Moreover, PRIMA-1 was tested on 34 human tumor cell lines of different tissues from the National Cancer Institute database and compared with five known anticancer drugs; PRIMA-1 was found to preferentially inhibit the growth of cell lines carrying *TP53* mutations as compared with cells expressing wild-type P53. In addition, a correlation between drug effectiveness and the levels of intracellular P53 mutant protein was found [53].

These initial observations generated some excitement, particularly considering that mutant P53 proteins, which escape the inhibitory effect of MDM2, accumulate in large amounts in malignant cells. Thus, the restoration of the wild-type functions to mutant P53 proteins by PRIMA-1 appeared capable of inducing massive apoptosis in tumor cells carrying mutant P53. Additional studies reported that PRIMA-1 treatment induced the expression of typical P53 target genes such as *P21*, *MDM2*, *PUMA*, NOXA, which contributed to the apoptosis induction of the cells carrying mutant P53 proteins from different tumors [54,55,56,57,58,59]. PRIMA-1 was also found to activate mitochondrial apoptosis through the release of cytochrome c [60], which could be independent of P53 transcriptional activity [61,62]. Besides PRIMA-1, also APR246 was shown to have a pro-apoptotic activity with higher efficiency. Moreover, PRIMA-1 and APR246 could act synergistically with different chemotherapeutic molecules to induce tumor cell apoptosis in vitro and to inhibit tumor xenograft growth in SCID mice in vivo [63]. The synergy between PRIMA-1 or APR246 with conventional antitumor drugs observed in several cell systems [64,65,66,67,68] indicated the potential advantages of a combination therapy capable of improving clinical efficacy, with a concomitant reduction of chemo-resistance and potentially detrimental side effects.

With regard to hematological malignancies, the effects of PRIMA-1, alone or in combination with fludarabine, were investigated in cells from CLL patients with or without del(17p) [69]. The *TP53* status was established only by FISH analysis, and data on the possible presence of *TP53* mutations were missing, although del(17p) can often be accompanied by a *TP53* mutation on the other allele [70]. Peripheral blood or bone marrow CLL cells were treated with PRIMA-1 and fludarabine alone or in combination; PRIMA-1 showed comparable cytotoxic effects on CLL cell samples with or without del(17p), while fludarabine was less effective in samples with del(17p). The treatment with both drugs in combination induced synergistic or additive effects in del(17p) cells, suggesting that PRIMA-1 enhanced the cytotoxicity of fludarabine in CLL cells with a P53 dysfunction [69]. These studies also showed that CLL cells were particularly susceptible to PRIMA-1 in contrast to normal lymphocytes that proved to be resistant to the same or even higher drug concentrations in vitro.

In contrast to the results obtained in CLL, PRIMA-1 reduced cell viability in AML cells with del(17p), whereas these cells were more resistant to conventional antileukemic drugs as Ara-C, chlorodeoxyadenosine, and fludarabine [71]. In a further study, APR246 proved to be pro-apoptotic and cytotoxic on cells from both mutant P53 AML cell lines and wild-type P53 primary AML patient cells in a dose- and time-dependent manner. These findings demonstrated that APR246 exerted its action irrespective of the *TP53* status. The synergistic effects of APR246 with conventional chemotherapeutic drugs observed on both wild-type and mutant P53 AML cells [72] confirmed that APR246 did not require the presence of a mutant P53 protein to exert its effects.

These data are consistent with the results by Lambert et al. on the molecular mechanisms through which PRIMA-1 and APR246 operate [73]. They found that both drugs are converted into the reactive compound methylene quinuclidinone (MQ), which covalently reacts with the thiol groups of both the mutant and the wild-type P53 proteins. The covalent modifications of the Cysteine (Cys) residues located in the core domain of the P53 protein are responsible for the restoration of a wild-type P53 conformation (Figure 1a). Computational analysis of structural models then identified a binding pocket containing Cys at positions 124, 135, and 141, between the L1 loop and S3 sheet in the P53 core domain, which could represent the possible targets of MQ, leading to the mutant P53 reactivation [74]. More recently, Cys277 of P53 has been identified as a prime binding target; specifically, Cys277 is essential for MQ-mediated thermostabilization of the core domains of the wild-type protein and R175H and R273H mutants, whereas Cys124 and Cys277 are both required for the reactivation of the R175H mutant mediated by APR246 [75].

These studies clarified how PRIMA-1 and APR246 work and demonstrated that the binding of MQ to P53 is capable of inducing the refolding of the mutant and of the wild-type misfolded proteins. Both these compounds can induce apoptosis in tumor cells, which carry mutant as well as wild-type P53, as observed in the case of AML [71] and MM [72,76] studies. Interestingly, Bao et al. demonstrated that PRIMA-1 and APR246 rescued inactive wild-type P53 in melanoma cells in which P53 was inhibited by a high level of α-v integrin signaling [77]. When tested in a 3-D collagen culture assay, APR246 activated P53 and caused the expression of the pro-apoptotic P53 target *APAF* and *PUMA* genes in several human melanoma cell lines. Furthermore, a P53-dependent inhibition of the growth of the melanoma cells treated with these drugs was observed in xenografted mice [77]. Therefore, PRIMA-1 and APR246 can conceivably be exploited to restore the function of the wild-type P53 protein inactivated by mechanisms other than mutations.

## 4. PRIMA-1 and APR246: The Second Phase, 10 Years Later

The effects of both PRIMA-1 and APR246 may be heterogeneous. For example, they may be unable to induce the expression of pro-apoptotic target genes in some mutant P53 expressing cells and to restore the binding of mutant P53 to DNA or to promote the transactivation of P53 target genes in certain experimental systems [64,78,79,80,81,82,83]. Collectively, these studies have indicated multiple mechanisms whereby these compounds can exert their effects in both mutant P53-dependent and -independent manners.

APR246 has the remarkable property of decreasing the cellular level of glutathione (GSH), while increasing that of ROS (Figure 1b) (Lambert et al., 2009). APR246 displayed significant cytotoxicity in primary MM cell samples with *TP53* locus deletion and in several cell lines with P53 alterations representative of the heterogeneity of those occurring in MM [i.e., del(17p), point mutations, exon deletions] by increasing intracellular ROS and depleting the GSH level [80]. In MM cells, the combined treatment with buthionine-sulfoximine (BSO), an irreversible inhibitor of γ-glutamyl cysteine-synthase (γ-GCS), induced a total depletion of GSH content and significantly synergized with APR246 in inducing cell death (Figure 1b). APR246 induced cell death also in MM primary cells, irrespective of their del(17p) status. Furthermore, it inhibited myeloma growth in a mouse xenograft model, and its efficacy increased when used in combination with BSO [80].

APR246 may also target and inhibit thioredoxin reductase 1 (TXNRD1), a key regulator of cellular redox balance (Figure 1b). This inhibition was assessed both in cell-free systems in vitro and in mutant P53 carrying cells treated with APR246 [84]. Knocking down the *TXNRD1* gene in cell lines with mutant P53 protein partially inhibited the induction of cell death caused by in vitro exposure to APR246; in these conditions, a decreased production of ROS in a P53-independent manner was also observed. Direct activity of the APR246 reactive molecule, MQ, on TXNRD1 could explain the observed cytotoxic effect on P53-null cells [84]. These findings have interesting implications for cancer therapy since APR246 might act synergistically with drugs targeting TXNRD1 to counteract the oxidative environment in tumor cells, thus inducing apoptosis. Indeed, Auranofin, an inhibitor of TXNRD1, synergizes with APR246, and is cytotoxic for MM cells in vitro irrespective of the *TP53* status [85].

Another piece of the “APR246 puzzle” has been identified in the NRF2 (nuclear factor erythroid 2-related factor)/HMOX1 (heme oxygenase 1) axis. In cells from the KMB3 AML cell line, APR246 was found capable of up-regulating genes of the heat shock and oxidative stress response, with *HMOX1* and *SLC7A11* (solute carrier family 7 member 11) genes being significantly up-regulated at the lowest drug dose [86]. The up-regulation of HMOX1 appeared independent from the *TP53* status but was dependent on NRF2 since the knocking-down of *NRF2* resulted in the suppression of *HMOX1* expression and increased cell death following APR246 treatment. This suggested that the activation of the NRF2/HMOX1 system, which represents a protective response triggered in parallel with ROS accumulation, was part of a cellular response to the treatment. This observation found further support when PI3K/mTOR inhibitors, such as wortmannin and rapamycin, were used in the experimental system in which the leukemic cells were exposed to APR246. The presence of one of the two drugs in vitro prevented the activation and nuclear translocation of NRF2 induced by APR246, together with the up-regulation of *HMOX1* expression, thus synergizing with APR246 in reducing cell viability [86]. Recently, Del Sal and colleagues using breast cancer cells have demonstrated that the mutant P53 protein may interact with NRF2 transcription factor acting as a switch to tune NRF2 activity and affecting the antioxidant response [87].

The relevance of the NRF2/SLC7A11 axis in affecting the GSH/ROS balance and consequently the APR246-induced cell death has been recently clarified in esophageal cancer cells carrying mutant or wild-type P53 and in patient-derived xenografts [88]. Specifically, MQ binds to thiol groups of GSH, leading to GSH depletion and ROS accumulation (Figure 1b). In cells with high levels of mutant P53, the mutant itself, by binding to NRF2, is able to suppress the expression of *SLC7A11*, the key component of the X_C_ system that imports cysteine necessary to the formation of GSH. In the presence of MQ, this leads to a further decrease of the GSH content and to an increase of ROS, causing APR246 cytotoxicity. Conversely, in wild-type or P53 null cells, *SLC7A11* expression remains high and the GSH may be maintained at sufficient levels to exert its functions, making the cells potentially resistant to APR246. Therefore, cells with high *SLC7A11* levels are expected to be resistant, while those with low *SLC7A11* expression to be sensitive to APR246, indicating SLC7A11 as a response predictor, as demonstrated in cells from the esophageal carcinoma and in cells from the NCI-60 cell lines panel, representing different solid and hematological malignancies [88].

PRIMA-1 and APR246 have also been shown to induce endoplasmic reticulum (ER) stress or unfolded protein response (UPR) in a panel of MM cell lines with different *TP53* functional status as well as in primary patient samples [82] (Figure 1c). Since PRIMA-1 cytotoxicity was higher in P53 null cells than in those with a mutant P53 protein, a P53-independent effect was postulated. Some UPR markers, such as *HSP70* and *GADD34*, were found significantly up-regulated by genome-wide gene expression analysis; an increase of P73, another member of the P53 family proteins, at both mRNA and protein level in the cells that were more sensitive to APR246 was also observed. Since P73 overexpression results in an enhancement of the expression of UPR markers, there was a clear indication of the involvement of UPR and ER stress in the cytotoxicity caused by APR246. Further proof for this hypothesis came from the observation that PRIMA-1 was capable of synergizing with bortezomib, a drug used for myeloma treatment and known to activate the UPR response [82] (Figure 1c). The involvement of UPR in APR246 induced cytotoxicity has also been highlighted in KMB3 AML cells, where the treatment with this drug induced the up-regulation of genes related to oxidative stress, ER stress, and UPR [86].

The involvement of P73 in APR246-mediated toxicity was also highlighted in primary human myeloma samples and cell lines as well as in clinically relevant xenograft models of MM, where the compound induced caspase-dependent apoptosis and inhibited colony formation regardless of the *TP53* status [76]. The induction of apoptosis was partially dependent on P53 activation but was rather related to the activation of P73 and its downstream target *NOXA*. Combined treatments of APR246 with dexamethasone or doxorubicin resulted in enhanced antitumor activity in vivo [76]. Furthermore, the treatment of MM cells with APR246 up-regulated miRNA-29a that, in turn, targeted c-Myc, a critical player in MM oncogenesis, for its down-regulation. Interestingly, intra-tumor delivery of miRNA-29a mimics induced tumor regression in mouse xenograft model of MM, synergizing with APR246 in this effect [89].

Autophagy also was investigated as an alternative cell death pathway induced by PRIMA-1, as reported in sarcoma [81], in breast and colon cancer cells [90]. In cell lines derived from soft-tissue sarcoma patients, APR246 induced autophagy regardless of the *TP53* status, and this correlated with an increase of ROS [81]. Autophagy was induced by PRIMA-1 in breast and colon cancer cells bearing both mutant and wild-type P53 as well as in their derivative P53-null cells; however, after treatment, a decrease of mutant but not of wild-type P53 protein was observed, suggesting that a protein degradation pathway, likely autophagy-related, acted specifically on mutant P53 protein [90]. PRIMA-1 cytotoxicity was shown to correlate with the degradation of mutant P53 in breast cancer cells, leading to speculate that, in these cells, the sensitivity to PRIMA-1 could depend on the abolition of GOF activity of the mutant P53, through a protein degradation pathway induced by the drug [91].

Recently, a similar activity of APR246 has been described in CLL cells [92]. In this study, APR246 efficiently induced cell death by apoptosis in both wild-type and mutant P53 bearing CLL cells; interestingly, in some samples with an accumulation of mutant P53 protein, APR246 significantly reduced P53 level, which correlated with lower viability and a more efficient apoptosis induction compared to samples carrying a stable mutant P53 [92]. Thus, this evidence, together with our previous observations, may suggest that both PRIMA-1 and APR246 are able to trigger protein degradation pathways directed to mutant P53 proteins that may reduce cancer cells’ survival (Figure 1d).

In conclusion, altogether, this evidence underlines a wide activity of PRIMA-1 and APR246 on different cellular pathways that can be both mutant P53-dependent and P53 status-independent and that likely may vary in a tissue-specific manner (Table 1).

## 5. Clinical Trials with APR246 in Hematological Malignancies

Ten years after PRIMA-1 isolation, its derivative APR246 has been tested in a phase I/II study in refractory hematologic malignancies and prostate cancers [93] (NCT 00900614) (Table 2). Patients affected by AML, CLL, and T-cell Prolymphocytic Leukemia (T-PLL) were enrolled in the study because leukemia has shown pronounced sensitivity in preclinical ex vivo studies. The primary objective of this study was to determine the maximum tolerated dose of the drug, its safety, and its pharmacokinetics, The 22 patients enrolled in this study received daily infusions of APR246 for four days. The analysis of cancer cells before and after treatment showed the induction of cell cycle arrest, apoptosis, and P53 target gene expression in several patients. Ten patients could be further evaluated, two of them showing signs of tumor regression [93]. The results suggest that APR246 may have a potential therapeutic effect.

A major promise for APR246 comes from the combination therapy with other drugs already used in chemotherapy (Table 2). Several trials are ongoing at different stages to study the safety and efficacy of the compound in combination with 5-azacitidine (5Aza) or venetoclax in AML and myelodysplastic syndrome (MDS) patients. 5Aza is a hypomethylating agent generally considered as the first-line treatment of MDS and AML patients with a *TP53* mutation. Low doses of APR246 alone or in combination with 5Aza reactivated the P53 pathway and induced an apoptotic program in *TP53*-mutated MDS and AML cell lines in vitro and in primary cells in vivo and in vitro. [27]. Indeed, data from two clinical Ib/II trials, NCT03072043 and NCT03588078, indicated a strong synergy of APR246 and 5Aza in MDS and AML patients carrying a mutant P53 protein, with the NCT03072043 study reporting an overall and complete remission rate of 86% and 53%, respectively [94]. Based on these data, the US Food and Drug Administration (FDA) granted Fast Track designation for the treatment of MDS patients carrying a mutant P53 protein with APR246 and 5Aza. Furthermore, a randomized Phase III study (NCT03745716) is ongoing in the treatment of *TP53*-mutated MDS patients to compare APR246 and 5Aza with 5Aza alone. Other active trials are in progress to determine the safety and the efficacy of APR246 in combination with 5Aza in AML or MDS patients following allogeneic stem cell transplant (NCT03931291) or in combination with Venetoclax in AML patients (NCT04214860).

Lastly, a study with the aim of determining the preliminary safety, tolerability, and pharmacokinetic profile of APR246 in combination with either Ibrutinib or Venetoclax plus rituximab therapy in NHL subjects with a mutant *TP53* status, also including CLL patients (NCT04419389), is about to begin.

## 6. Conclusions and Perspectives

PRIMA-1 and APR246 can be considered relevant molecules with an antitumor effect in different cancer types. They are able to induce the up-regulation of genes involved in cell cycle control and apoptosis in mutant P53 bearing cancer cells, leading to the induction of apoptotic cell death. However, both molecules have additional effects, which are independent from the P53 transcriptional activity or from the *TP53* status of the gene, such as those involving the oxidative and ER stress (Figure 1b,c). These are likely to increase the therapeutic efficacy and potentially widen the clinical uses of these compounds on tumor cells. Specifically, combination therapies making use of APR246 together with molecules acting on redox regulation (such as Auranofin and BSO) or unfolded protein response (such as Bortezomib) might open new therapeutic opportunities. The exploitation of these additional features of the compound is nevertheless quite demanding both in terms of preclinical efforts and also for the design of ad hoc clinical trials.

The potential activities of PRIMA-1 and APR246 have been extensively studied in AML and MM among the hematological malignancies while remaining quite unexplored in CLL. Although APR246 has received breakthrough therapy, orphan drug and fast track designations from the FDA for MDS, and Orphan Drug designation from the European Commission for MDS, AML, and ovarian cancer, it has not yet been approved by the FDA or any regulatory authority. Moreover, the potential of APR246 has not been specifically investigated in the CLL setting, possibly due to the combination of two factors, comprising the availability of several new treatment options and the requirement for additional preclinical testing with APR246. However, since targeted therapies in CLL aim at blocking the two main pathways that promote disease progression (i.e., BCR stimulation and apoptosis) [95], it would be interesting to clinically test the added value of the functional restoration of the P53 protein by APR246. The issues of the relationship between the heterogeneous functionality of P53 mutants, together with the consideration that CLL is characterized by the occurrence of different *TP53* mutations with potential different responses to P53-targeted drugs, should also be addressed. Moreover, mutant P53 proteins show different degrees of functionality in CLL, and it is plausible that patients expressing a P53 mutant protein with partial activity might show a different response to APR246 treatment compared to patients carrying a completely inactive P53 protein. On the other hand, other activities of APR246, that may impact on the cellular redox balance, the glucose metabolism, or the protein degradation pathways (i.e., autophagy) could effectively cause the death of CLL cells, characterized by altered lipid metabolism, mitochondrial activity, and reactive oxygen species formation [96]. As an example, the recent report from Jaskova et al. suggests that APR246 can activate alternative pathways in CLL that promote the elimination of mutant P53 protein and increase cell death [92]. How all these activities are related to the specific functionality of CLL mutated P53 proteins still remains an open issue; indeed, further studies on the functional characterization of different P53 proteins expressed in CLL cells as well as the results obtained in the recently activated clinical trial will help to shed light on the role of P53 and its mutant proteins in CLL pathogenesis.

## Figures and Tables

**Figure 1 cells-10-00098-f001:**
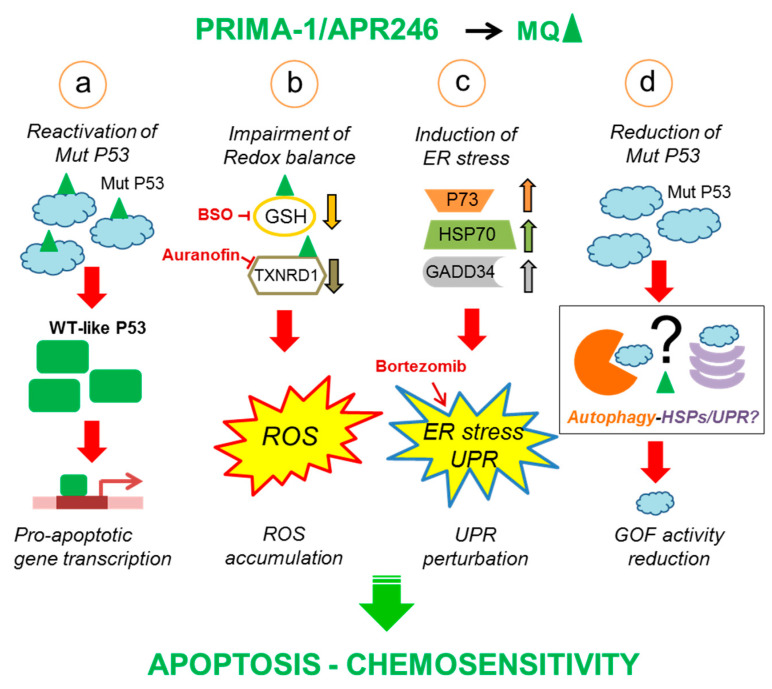
PRIMA-1/APR246 activities reported in hematological tumor cells. The main activities of PRIMA-1/APR246 leading to apoptosis induction and cell death are depicted: (**a**) the reactivation of mutant P53, leading to the expression of P53 target genes and apoptosis induction; (**b**) the perturbation of redox balance mediated by a reduction of GSH level and TXNRD1 activity due to direct binding of the PRIMA-1/APR246 reactive compound MQ; molecules that can further impair GSH and TXNRD1 activity (BSO and Auranofin, respectively) are indicated; (**c**) the induction of ER stress and UPR mediated by upregulation of HSP70, GADD34, and P73; bortezomib able to enhance the UPR and synergize with APR246 to increase tumor cell death is indicated; (**d**) the reduction of mutant P53 levels through unknown pathways (black box) that may result in the decrease of mutant P53 GOF activity. ROS, reactive oxygen species; HSPs, heat shock proteins; UPR, unfolded protein response; ER, endoplasmic reticulum; Mut P53 (mutant P53); WT (wild-type).

**Table 1 cells-10-00098-t001:** Mutant P53-dependent and -independent activities of PRIMA-1/APR246 reported in hematological tumor cells.

**Mutant P53-Dependent**	**Cancer Type**	**Ref**
Reactivation of wild-type P53 functions	BL, AML, MDS	[18,27,59,73]
ROS induction	BL, Hemat. Tumors	[73,88]
Reduction of mutant P53 level	CLL	[92]
**P53 status-independent**	**Cancer Type**	**Ref**
Impairement of GSH/ROS balance	MM, BL, AML	[80,84,85,86,88]
Induction of ER stress and UPR	MM, AML	[76,82,86]

BL, Burkitt lymphoma; MDS, myelodysplastic syndrome; AML, acute myeloid leukemia; MM, multiple myeloma; CLL, chronic lymphocytic leukemia; GSH, glutathione; ROS, reactive oxygen species; ER, endoplasmic reticulum; UPR, unfolded protein response.

**Table 2 cells-10-00098-t002:** Clinical trials with APR-246 in hematological malignancies.

Trial ID	Phase	Objectives	Conditions	Drugs	Status
NCT 00900614	I	To determine the highest feasible dose of intravenous APR246 when given to patients with refractory hematologic malignancies or prostate carcinoma.	AML, CLL, TPLL, NHL, MM, Prostatic Neoplasms	APR246	Completed
NCT 03072043	Ib/II	To determine the safe and recommended dose of APR246 in combination with 5-Azacitidine as well as to see if this combination of therapy improves overall survival.	MDS, AML, MPN, CMML	APR246 5Aza	Active
NCT 03588078	Ib/II	To evaluate the safety and efficacy of APR246 in combination with 5-Azacitidine for the treatment of *TP53* Mutant Myeloid Neoplasms as well as to see the CR of this patients	MDS With Gene Mutation; AML With Gene Mutations; MPN, CMML	APR246 5Aza	Active
NCT 03745716	III	To compare the rate and duration of CR, in patients with *TP53*-mutated MDS who will receive APR246 and Azacitidine or 5-Azacitidine alone.	MDS, AML	APR246 ± 5Aza	Active
NCT 03931291	II	To assess the safety and efficacy of APR246 in combination with azacitidine as maintenance therapy after allogeneic HSCT (hematopoietic stem cell transplant) for patients with *TP53* mutant AML or MDS.	AML, MDS	APR246 ± 5Aza	Active
NCT 04214860	I	To determine the safety and preliminary efficacy of APR246 in combination with Venetoclax and Azacitidine in patients with myeloid malignancies.	AML	APR246 ± Venetoclax; APR246 ± 5Aza	Active
NCT044119389	I/II	Study to determine the preliminary safety, tolerability, and pharmacokinetic profile of APR246 in combination with either ibrutinib or venetoclax + rituximab therapy in subjects with *TP53*-mutant NHL, including R/R CLL and R/R MCL.	NHL, CLL, MCL	APR246 ± Ibrutinib; APR246 ± Venetoclax-R	Recruiting

From: https://www.clinical.trials.gov; AML, acute myeloid leukemia; CLL, chronic lymphocytic leukemia; TPLL, T-prolymphocytic leukemia; NHL, non-Hodgkin lymphoma; MM, multiple myeloma; MDS, myelodysplastic syndrome; MPN, myeloproliferative neoplasm; CMML, chronic myelomonocytic leukemia; MCL, mantle cell lymphoma; CR, complete response; R/R relapsed and/or refractory.

## Data Availability

Data sharing not applicable.

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
