# Peer review of "Antitumor Effects of PRIMA-1 and PRIMA-1^Met^ (APR246) in Hematological Malignancies: Still a Mutant P53-Dependent Affair?"

_cells, 2021, doi:10.3390/cells10010098_

Round 1
Reviewer 1 Report
In this review, the authors introduced mutant P53-dependent and –independent antitumor effects of PRIMA-1 and PRIMA-1Met (APR246) and described their potential use in hematological malignancies. To my impression, the review is presented in a well-organized and logical manner. All the experimental results show reasonable consistency. In addition, these studies provide insightful knowledge of PRIMA-1/APR-246 and will contribute to further studies on its applications. I would therefore strongly recommend this review for publication in Cells.
Author Response
Response to Reviewer #1:
In this review, the authors introduced mutant P53-dependent and –independent antitumor effects of PRIMA-1 and PRIMA-1Met (APR246) and described their potential use in hematological malignancies. To my impression, the review is presented in a well-organized and logical manner. All the experimental results show reasonable consistency. In addition, these studies provide insightful knowledge of PRIMA-1/APR-246 and will contribute to further studies on its applications. I would therefore strongly recommend this review for publication in Cells.
It was not clear to us the point regarding the English since the reviewer required an extensive editing of English language and style, but gives 5 stars at the question: Is the English used correct and readable. However, a revision of the English has been made.
Reviewer 2 Report
It has been 18 years since the publication of the seminal article on Prima-1. Since then, a number of publications have focused on mutant P53-dependent antitumor effects of PRIMA-1 and PRIMA-1Met. In my understanding, once it was demonstrated that PRIMA-1 targets sulfhydryl groups, the whole concept that PRIMA-1 specifically hits mutant P53 was severely damaged. The fact that 18 years after the initial publication there are only a bunch of phase I-II clinical trials, and only one phase III study, is in line with the biochemical considerations. Basically this review article is an insigthful critical narrative of the PRIMA-1 story. My sole concern is about the title that does not fully reflect the efforts made by the authors to achieve a certain distance permitting a fair level of analysis. I would suggest to reword the title in a way (a bit more provocative?) that makes it distinct from the mainstream literature.
Author Response
Response to Reviewer #2:
It has been 18 years since the publication of the seminal article on Prima-1. Since then, a number of publications have focused on mutant P53-dependent antitumor effects of PRIMA-1 and PRIMA-1Met. In my understanding, once it was demonstrated that PRIMA-1 targets sulfhydryl groups, the whole concept that PRIMA-1 specifically hits mutant P53 was severely damaged. The fact that 18 years after the initial publication there are only a bunch of phase I-II clinical trials, and only one phase III study, is in line with the biochemical considerations. Basically this review article is an insigthful critical narrative of the PRIMA-1 story. My sole concern is about the title that does not fully reflect the efforts made by the authors to achieve a certain distance permitting a fair level of analysis. I would suggest to reword the title in a way (a bit more provocative?) that makes it distinct from the mainstream literature.
As suggested, we reworded the title in this way:
- i) Antitumor effects of PRIMA-1 and PRIMA-1Met (APR246) in hematological malignancies: still a mutant P53-dependent affair?